# Using hypergraphs to quantify importance of sets of diseases by healthcare resource utilisation: A retrospective cohort study

James Rafferty[1]*, Alexandra Lee[1], Ronan A. Lyons[1], Ashley Akbari[1], Niels Peek[2,3], Farideh Jalali-najafabadi[4], Thamer Ba Dhafari[2], Jane Lyons[1], Alan Watkins[1], Rowena Bailey[1]

**1** Population Data Science, Swansea University Medical School, Swansea University, Swansea, United Kingdom, **2** Division of Informatics, Imaging and Data Science, School of Health Sciences, The University of Manchester, Manchester, United Kingdom, **3** Alan Turing Institute, London, United Kingdom, **4** Centre for Genetics and Genomics Versus Arthritis, Centre for Musculoskeletal Research, Faculty of Biology, Medicine and Health, Manchester Academic Health Science Centre, The University of Manchester, Manchester, United Kingdom

* j.m.rafferty@swansea.ac.uk

**Data Availability Statement:** This study makes use of anonymized, individual-level data held in the SAIL Databank, a Trusted Research Environment,

## Abstract

Rates of Multimorbidity (also called Multiple Long Term Conditions, MLTC) are increasing in many developed nations. People with multimorbidity experience poorer outcomes and require more healthcare intervention. Grouping of conditions by health service utilisation is poorly researched. The study population consisted of a cohort of people living in Wales, UK aged 20 years or older in 2000 who were followed up until the end of 2017. Multimorbidity clusters by prevalence and healthcare resource use (HRU) were modelled using hypergraphs, mathematical objects relating diseases via links which can connect any number of diseases, thus capturing information about sets of diseases of any size. The cohort included 2,178,938 people. The most prevalent diseases were hypertension (13.3%), diabetes (6.9%), depression (6.7%) and chronic obstructive pulmonary disease (5.9%). The most important sets of diseases when considering prevalence generally contained a small number of diseases, while the most important sets of diseases when considering HRU were sets containing many diseases. The most important set of diseases taking prevalence and HRU into account was diabetes & hypertension and this combined measure of importance featured hypertension most often in the most important sets of diseases. We have used a single approach to find the most important sets of diseases based on co-occurrence and HRU measures, demonstrating the flexibility of the hypergraph approach. Hypertension, the most important single disease, is silent, underdiagnosed and increases the risk of life threatening co-morbidities. Co-occurrence of endocrine and cardiovascular diseases was common in the most important sets. Combining measures of prevalence with HRU provides insights which would be helpful for those planning and delivering services.

at Swansea University, Swansea, UK. Due to the nature and level of the data, data are not publicly available but are available to researchers upon application. All proposals to use SAIL data are subject to review by the independent Information Governance Review Panel (IGRP). The IGRP gives careful consideration to each project proposal to ensure appropriate use of SAIL data. If a project is approved, access to the requested data is gained through a privacy-protecting safe haven and remote access system referred to as the SAIL Gateway. SAIL has established an application process to be followed by anyone who would like to access data at https://saildatabank.com/sail_user_application/ and further information is available by emailing SAILDatabank@swansea.ac.uk.

**Funding:** This work was supported by the Medical Research Council (MRC), grant no. MR/S027750/1. FJ is supported by a MRC/University of Manchester Skills Development Fellowship (grant number MR/R016615). The funders had no role in study design, data collection and analysis, decision to publish, or preparation of the manuscript.

**Competing interests:** The authors have declared that no competing interests exist.

## Introduction

Multi-morbidity, or multiple long term conditions (MLTC) is defined as the presence of two or more long-term conditions [1]. The prevalence of multi-morbidity is increasing across the world due to ageing populations and improved survival for many chronic conditions [2–4]. multi-morbidity is also more common in less affluent or educated communities [5, 6]. Historically, health research has generally focused on single diseases, so comparatively little is known about how multiple diseases and treatments interact. Little is known about which combinations of conditions are most prevalent or troublesome. Our aim was to quantitatively evaluate combinations of long term conditions to determine their importance when considering prevalence and also their impact on healthcare resource utilisation, for which we used the standardised rate at which people interacted with outpatient services or as unplanned inpatient services admission. Understanding such resource utilisation would be valuable for better planning of healthcare and improving patient outcomes.

Several studies investigating how multi-morbidity affects patient outcomes using linear or time-to-event regression, such as Cox regression, exist in the literature. For example, [7] developed a prediction model to estimate the risk of additional chronic diseases using a copula-based approach. [8, 9] used multiple logistic regression to model the relationship between multi-morbidity and some outcome measure. [10, 11] used Cox regression methods to analyse the interplay between multi-morbidity and long term mortality. Several groups have recently used machine learning models, typically random forests, to investigate how multi-morbidity relates to certain outcomes, for example, random forests were used by [12] to develop a multi-morbidity frailty index and by [13] to investigate the relationship between multi-morbidity and healthcare expenditure. Deep learning models have recently been applied to the problem of healthcare resource utilisation. For example [14] used an attention-based model to predict operations from diagnosis data in secondary care, while [15] investigated predicting healthcare expenditure from multiple sources of input data.

The use of statistical modelling and unsupervised machine learning techniques to find clusters of coincident diseases and understand multi-morbidity has been thoroughly explored in the literature recently [16]. Several groups observed that network analysis could be used to describe a system of diseases and the interactions between pairs of disease (see for example [17–23]).

Network based approaches utilising mathematical structures called graphs and hypergraphs have several advantages over other clustering and statistical modelling approaches. Many clustering methods (for example, hierarchical clustering) allow the user to only include each disease in a single cluster, which may obscure interactions where a single disease is an important feature of several disease clusters. Approaches utilising weighted graphs are useful since they allow us to account for the prevalence of individual conditions while also accounting for the number of people that have different combinations of diseases (which can be thought of as a measure of 'prevalence' of sets of diseases).

We used a hypergraph rather than a simpler binary graph for this work because hypergraphs can quantify the effect of interactions between any number of diseases (as opposed to just two in a binary graph). The calculation of a quantity from the graph called centrality allows one to quantitatively estimate the connectedness of nodes within the graph. Nodes with a high centrality are strongly connected to other nodes, which for a graph where nodes represent diseases and the edge weights represent the number of people that have all diseases connected to the edge, represents the frequency at which a single disease features in sets of diseases (and hence, it's importance to multi-morbidity). We used a network approach and a hypergraph in a previous original article and tutorial paper [24]. Subsequently, work using hypergraphs has been performed by others [25].

The measure of prevalence of sets of diseases is only one choice of weighting scheme. Any measure related to the sets of diseases can be used to weight the hypergraph, which reflects the flexibility of the approach. In this study we have chosen to use healthcare resource utilisation as the quantity with which we will weight the graph, since healthcare resource use is an important factor to understand for healthcare delivery planning purposes, and is likely to be highly correlated with negative patient outcomes.

This aims of this work were

1. To demonstrate the utility of hypergraphs in their application to problems of quantifying disease set importance based on healthcare resource use, a metric unrelated to the prevalence of the diseases.

2. To find the most important sets of diseases based on two different measures of healthcare resource use, interactions with outpatient services and unplanned interactions with inpatient services, and a similar measure of prevalence to that used in previous work [24] making a total of three weightings for consideration.

## Methods

This study was performed using anonymised routine data held in the Secure Anonymised Information Linkage (SAIL) Databank, a Trusted Research Environment for people interacting with healthcare services in Wales, UK. The study was approved by the Information Governance Review Panel under project reference number 0911. Ethical approval was not required nor sought for the study.

### Cohort

The cohort used for this study consisted of all people living in Wales, UK on the 1st January 2000 and aged 20 years or older, which was constructed specifically to study multimorbidity in Wales, UK. Please see [26] for a full description of the cohort used. All clinical events recorded in primary or secondary care before the index date of 1ˢᵗ January 2015 were included in the study. The raw data consisting of Read coded primary care data and ICD-10 coded secondary care data were processed into a table containing one row per pseudonymised person ID. The outcome measures chosen were the standardised rate of unplanned admissions to inpatient care and the standardised rate of interactions with outpatient services recorded in the three years following the index date (i.e. from 1ˢᵗ January 2015 to 31ˢᵗ December 2017). All data for the study were accessed and analysed within the SAIL Databank [27, 28].

### Hypergraph data

Hypergraphs were constructed using software available from [29]. Three hypergraphs were built separately using different weighting schemes to quantify diseases prevalence and healthcare resource utilisation. In each hypergraph diseases were represented by nodes and sets of diseases by edges [24]. The disease data were derived from primary and secondary care records held in the Welsh Longitudinal General Practice Database and the Patient Episodes Database for Wales respectively, and consisted of binary flags indicating the presence or absence of specific disease diagnoses in patient records. The disease definitions used were taken from the Elixhauser morbidity index [30, 31]. Three pairs of diseases were merged (i. cancer and metastatic cancer, ii. diabetes and diabetes with complication, iii. hypertension and hypertension with complication) since they are closely related by nature and would induce pseudoclustering.

Firstly for the prevalence hypergraph node weights, we used the prevalence of the diseases

$$w_i^N = \frac{|X_i|}{P}$$

where $|X_i|$ is the number of people with disease $X_i$ and $P$ is the total population. For the edge weights we chose the generalised overlap coefficient

$$w_a^E = \frac{|X_i \cap X_j \cap X_k \cap \ldots \cap X_l|}{\min\left(|X_i|, |X_j|, |X_k|, \ldots, |X_l|\right)}$$

where $E_a = \{X_i, X_j, X_k, \ldots, X_l\}$.

For the outpatient resource utilisation hypergraph the node and edge weights were the age standardised rate of interactions with outpatient services per 100,000 people recorded for people with the specific disease or set of diseases (i.e., the edge weight for the diabetes and rheumatoid arthritis edge used only people with recorded diagnoses of diabetes, rheumatoid arthritis and no other diseases under consideration in the study but we note that people may have recorded diagnoses for diseases that were not considered as part of the study). For the unplanned inpatient resource utilisation hypergraph the weighting scheme was the age standardised number of unplanned admissions to inpatient care recorded for people with the specific disease or set of diseases. Age standardisation was performed using the European standard population 2013 [32].

We then computed the eigenvector centrality of the dual representation of each hypergraph. The eigenvector centrality of the nodes and edges of a hypergraph give a direct measure of the importance of each to the graph as a whole and as such are interpreted as the importances of the diseases and sets of diseases. Uncertainties were calculated using bootstrapping, i.e. a bootstrap cohort was selected from main cohort with replacement and the hypergraph and eigenvector centrality were calculated, and this process was repeated. The mean of the eigenvector centrality for each set of diseases was taken as the estimate of centrality, while the 2.5% and 97.5% percentiles of the bootstrap distribution were taken as the 95% confidence intervals. We discarded sets from the results where the lower 95% confidence interval of the centrality intersected with zero.

## Further analysis

In order to construct a picture of sets of diseases that are important both because of their prevalence and because of their healthcare resource utilisation we constructed a composite measure of these quantities. Firstly, we found the sets of diseases that had a centrality higher than the median for both hypergraphs. We then found the Euclidean sum of the eigenvector centralities (i.e. the square root of the sum of the squares of the individual eigenvector centralities). We constructed three composite measures, two combining the overlap coefficient centralities with the HRU centraties to investigate differences in care needs for different sets of diseases and finally one combined measure of all three centralities.

## Results

Data from a total of 2,178,938 people were included in the analysis. See Table 1 for summary statistics. The most commonly diagnosed diseases were hypertension (13.3%), diabetes (6.9%), depression (6.7%) and COPD (5.9%). The frequency of the number of interactions with healthcare services for both outpatient and unplanned inpatient services was approximately exponential other than an enhanced zero count, as can be seen in Fig 1.

**Table 1. The fraction of people diagnosed with the feature diseases.**

| Disease | Diagnosed (%) |
|---|---|
| Hypertension | 13.30 |
| Diabetes | 6.88 |
| Depression | 6.69 |
| COPD | 5.89 |
| Any Cancer | 4.96 |
| Renal Disease | 4.83 |
| Obesity | 4.56 |
| Arrythmia | 4.52 |
| Other Neurological Disorders | 3.61 |
| Hypothyroidism | 2.77 |
| Deficiency Anaemia | 2.58 |
| Congestive heart failure | 2.06 |
| Fluid & Electrolyte Disorders | 1.98 |
| Weight Loss | 1.87 |
| Valvular Disease | 1.69 |
| Peripheral Vascular Disease | 1.61 |
| Rheumatoid Arthritis | 1.49 |
| Peptic Ulcer | 0.94 |
| Liver Disease | 0.73 |
| Pulmonary Circulation Disorder | 0.67 |
| Paralysis | 0.54 |
| Drug Abuse | 0.53 |
| Psychosis | 0.47 |
| Coagulopathy | 0.34 |
| Lymphoma | 0.22 |
| Blood loss anaemia | 0.12 |

The most important sets of diseases by prevalence all featured hypertension and were generally smaller sets, with the most important being hypertension and diabetes. The most important set containing three conditions and the 20th most important set overall was hypertension, diabetes and obesity (see supplement for the 100 most important disease sets). the most

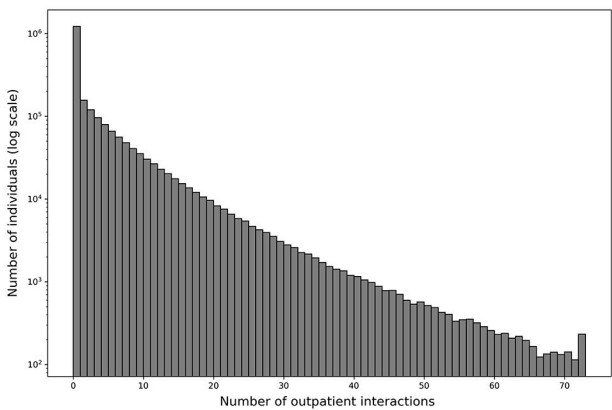 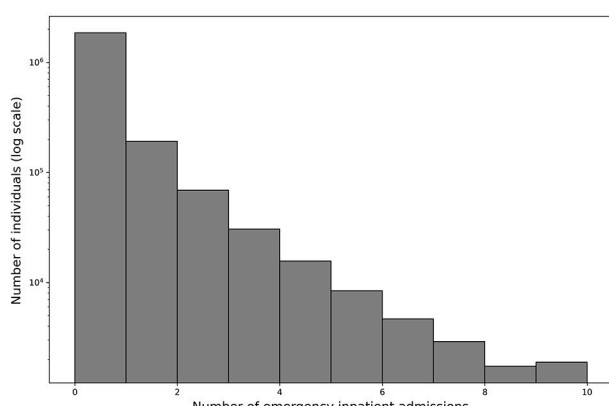

**Fig 1. The frequency of the number of interactions with healthcare services.** The y-axes are logarithmic scales. Left: Outpatient services. Right: Unplanned inpatient services.

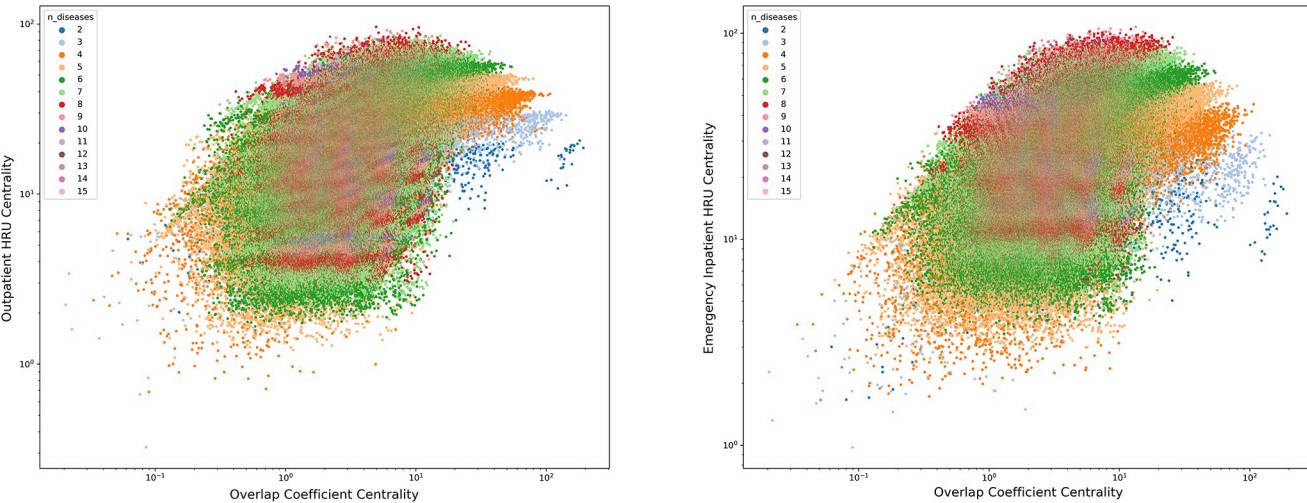

**Fig 2. The centrality of the overlap coefficient weighted hypergraph (x-axis) plotted against the centrality of the HRU hypergraph (y-axis).** Each point represents a set of diseases. The colour of the point represents the number of diseases in the set. Left: Outpatient services. Right: Unplanned inpatient services.

important set of diseases for unplanned inpatient HRU was a set of nine diseases (Arrhythmia, COPD, heart failure, fluid and electrolyte disorder, peripheral vascular disease, pulmonary circulatory disorder, renal disease, valvular disease and hypertension). The most important set of diseases for outpatient HRU contained eight diseases (COPD, heart failure, depression, fluid and electrolyte disorder, obesity, peripheral vascular disease, renal disease and valvular disease) (see supplements for the 100 most important disease sets for both unplanned inpatient and outpatient HRU).

When combined to investigate sets of diseases that were important for both prevalence and one of the HRU hypergraphs, we found the most important set of diseases for overlap coefficient combined with unplanned inpatient HRU was arrhythmia, heart failure and hypertension while for overlap coefficient combined with outpatient HRU the most important set was diabetes and hypertension. See Fig 2 for a plot of the overlap coefficient hypergraph centrality against the HRU weighted hypergraph centrality and supplemental material for tables of the 100 most important sets. In Fig 2, each point represents a set of diseases, and the distance from the origin of each point represents the combined importance of the set of diseases. The colour of the points represents the number of diseases in the set. It is evident that larger sets of diseases typically have larger HRU centrality values, while smaller sets of diseases typically have larger overlap coefficient centrality values.

When all three hypergraph centralities were combined, the most important set of diseases was diabetes and hypertension. All of the top 17 sets of diseases featured hypertension. See the supplemental material for a table of the 100 most important sets.

The single diseases that were included most often in the most important sets of diseases were hypertension, appearing in 60.1% of the most important disease sets for outpatient HRU and 59.9% of the most important disease sets for unplanned inpatient HRU, followed by arrhythmia and renal disease (see Fig 3).

## Discussion

This work demonstrates the use of hypergraph analysis for applied multi-morbidity research beyond simply describing clusters of coincident diseases. The weights in a hypergraph can in

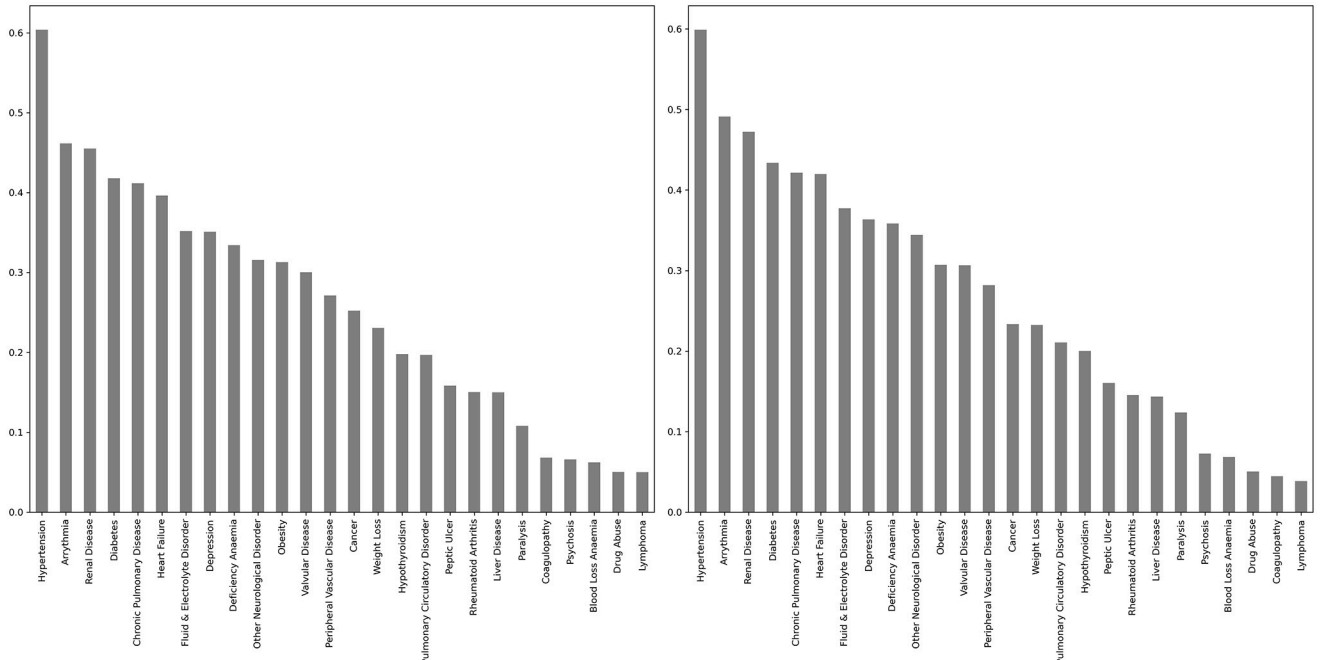

**Fig 3. The number of times each single disease appeared in the most important sets of diseases based on combined centrality for overlap coefficient and HRU weighted hypergraphs.** Left: Outpatient services. Right: Unplanned inpatient services.

principle be used to quantify any relationship between the nodes of the hypergraph which makes them supremely flexible and useful mathematical objects for modelling many things, including differences in healthcare resource utilisation between people that have different sets of diseases.

The importance of the sets of diseases when the hypergraphs were considered on their own exhibited patterns one would expect (see for example [33] which had similar findings to our study). There is currently no accepted standard method for defining the diseases that are considered in studies of multimorbidity, and considerable variation in methods used in the literature has led to different sets of diseases and corresponding differences in conclusions [34].

One needs to be careful when interpreting the results of hypergraph centrality, especially when the weighting scheme is more abstract like a measure of HRU. Eigenvector centrality, as used here, is high when a node is strongly connected to other nodes that also exhibit high centrality. For hypergraphs with weightings that depend on a measure of prevalence an important disease set is one where the number of people with the set of conditions is relatively high compared to other disease sets in the hypergraph, and also that the disease set is strongly connected to other disease sets. This implies that people with a specific 'important' set of diseases are more likely to acquire new diseases. When the weighting scheme used to construct the hypergraph is more abstract, such as HRU for each set of diseases, the interpretation is more complicated. A high centrality means that the HRU of the set of diseases is large, but the HRU of neighbouring sets of diseases is also large. This means that if a person has an 'important' set of conditions, then removing or adding a disease from the set wouldn't have a large effect on the approximate HRU.

We chose to consider importance of multi-morbidity using two axes, prevalence and HRU. The most important sets of diseases for the prevalence hypergraphs were sets containing few diseases, because the number of people that had a small number of diseases was large

compared to the number that had many diseases. The most important set of diseases in the prevalence hypergraph was hypertension with diabetes, both of which are very prevalent conditions.

Conversely, the most important sets of diseases for the hypergraphs weighted by HRU consisted of many diseases, which is also natural as people with many diseases are likely to have more complex healthcare needs that will require involvement of clinicians from different specialties. Despite the centrality being large for many large sets of diseases the confidence interval around the centrality depended on the number of people in the set and often became very large when the number of diseases in the set was large. We discarded sets where the lower 95% confidence interval of the centrality intersected with zero. This had the effect of removing sets of diseases where the centrality was indistinguishable from zero, but also had the effect of removing sets with small numbers of individuals in them.

Arguably, the most important sets of multi-morbidities are those which are both prevalent and command a large HRU. To find these sets we combined most important sets of diseases from two hypergraphs, the prevalence centrality combined with a measure of HRU centrality. This created a different set of rankings. The most important sets of conditions for the prevalence and unplanned inpatient combination was arrhythmia, heart failure and hypertension while for the combination of prevalence and outpatients HRU was diabetes and hypertension. We observed that the most important sets of diseases for the outpatient activity combination were smaller than for the emergency inpatients combination. Hypertension appeared prominently in the list of most important sets for both combinations.

Combining centrality measures for all three hypergraphs we have computed provides an overall picture of the diseases that have the highest general HRU and are relatively prevalent compared to other sets of diseases. The most important sets of diseases using this measure all featured hypertension, with the most important set being diabetes and hypertension. For future work one may consider applying a weighting to the combination of centrality measures. In this study the hypergraph centralities are all weighted equally, but we note that when combining a prevalence hypergraph and a HRU hypergraph the prevalence component contributes 50% of the combined centrality, but for this combination of all three hypergraphs the prevalence component only contributes 33.3% of the combined centrality. The contribution of hypergraph weights were allocated equally since there was no identified a-priori method to allocate disproportionate weightings. This observation could be used to tailor the method to specific research questions, for example, in demographic groups where the majority of healthcare interactions are delivered via outpatient services, the hypergraph derived from outpatient HRU could be weighted more highly in the combined centrality than inpatient HRU centrality.

Hypertension was the disease that appeared most frequently in sets of diseases that were important in both the prevalence and HRU hypergraphs by a large margin. Hypertension is a "silent" condition, inasmuch as moderate or even severe hypertension often presents with no symptoms. Furthermore, it is commonly associated with an increased risk of life-threatening cardiovascular conditions like heart attack and stroke. The order of the single diseases that appear most commonly in the most important sets of diseases is largely the same for the two measures of HRU, meaning people who have higher HRU for unplanned impatient care are likely to also have higher HRU for outpatient services.

This study has presented a combined analysis of disease set prevalence and HRU using hypergraphs. We have quantitatively evaluated sets of diseases based on their prevalence and their HRU and ordered the sets of diseases based on importance. The study has the advantage of providing a quantitative estimate for the importance of every set of diseases (some methods for clustering diseases require that diseases can only appear in one set for example) and

hypergraph objects are general enough to allow one to choose the weighting scheme used to capture the information needed by the research. A limitation of the hypergraph approach is they it can be very time consuming to compute, as the number of edges scales exponentially with the number of nodes. This makes bootstrapping to calculate uncertainties quite time consuming, even on a computing cluster. The flexibility to define the hypergraph weights also leads to some difficulties in the interpretation of hypergraph centrality.

Our results are coherent on the relatively small but growing literature on the impacts of multimorbidity. Soley-bori and colleagues carried out a systematic review of the impact of multimorbidity on healthcare costs and utilisation in the UK [35] in 2020, identifying 17 studies (7 on costs and 10 on HRU). Whilst the different studies used different demographic inclusion criteria, grouping of morbidity, and time frames the overall patterns were similar; multimorbidity found to be associated with increased primary care, emergency department and inpatient resources. Similar patterns have been reported from Denmark, India, Catalonia and China, again using different categories and methodologies [36–39].

The results of this study should be of interest to health planners, patients and patient advocacy groups. Combining measures of prevalence with HRU provides insights into aspects of the 'importance' of sets of multi-morbidities which would be helpful for those planning services. For future work it may be of interest to perform this analysis in cohorts of people with specific, common diseases to understand the common sets of comorbidities and HRU in those subcohorts. From this study, the most interesting populations to explore in studies of this type would be people with hypertension, diabetes or depression.

## Supporting information

**S1 File. Prevalence hypergraph most important disease sets.** The one hundred most central sets of diseases based on prevalence weighting.
(CSV)

**S2 File. Unplanned inpatients hypergraph most important disease sets.** The one hundred most central disease sets from a hypergraph weighted using the number of unplanned inpatient visits.
(CSV)

**S3 File. Outpatients hypergraph most important disease sets.** The one hundred most central disease sets from a hypergraph weighted using the number of outpatient visits.
(CSV)

**S4 File. Prevalence and outpatients HRU most important disease sets.** The one hundred most important diseases from the prevalence and outpatients HRU hypergraphs combined into a single importance score.
(CSV)

**S5 File. Prevalence and unplanned inpatients HRU most important disease sets.** The one hundred most important diseases from the prevalence and unplanned inpatients HRU hypergraphs combined into a single importance score.
(CSV)

**S6 File. Prevalence, unplanned inpatient and outpatients HRU most important disease sets.** The one hundred most important diseases from the unplanned inpatient, prevalence and outpatients HRU hypergraphs combined into a single importance score.
(CSV)

## Acknowledgments

This study makes use of anonymised data held in the SAIL Databank, which is part of the national e-health records research infrastructure for Wales. We would like to acknowledge all the data providers who make anonymised data available for research.

## Author Contributions

**Data curation:** Jane Lyons.

**Formal analysis:** James Rafferty.

**Funding acquisition:** Ronan A. Lyons, Niels Peek.

**Methodology:** James Rafferty.

**Project administration:** Rowena Bailey.

**Software:** James Rafferty, Alexandra Lee.

**Supervision:** Rowena Bailey.

**Writing – original draft:** James Rafferty.

**Writing – review & editing:** James Rafferty, Alexandra Lee, Ronan A. Lyons, Ashley Akbari, Niels Peek, Farideh Jalali-najafabadi, Thamer Ba Dhafari, Jane Lyons, Alan Watkins, Rowena Bailey.

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
