## [Decision Letter · Decision Letter 0]

15 Nov 2022

PONE-D-22-20775Using hypergraphs to quantify importance of sets of diseases by healthcare resource utilisation: A retrospective cohort studyPLOS ONE

Dear Dr. Rafferty,

Thank you for submitting your manuscript to PLOS ONE. After careful consideration, we feel that it has merit but does not fully meet PLOS ONE’s publication criteria as it currently stands. Therefore, we invite you to submit a revised version of the manuscript that addresses the points raised during the review process.

ACADEMIC EDITOR:Please share in the background section to show that this research is essential and has a specificity that distinguishes it from previous research.

Added an explanation in the method section.

Improve the writing of results and deepen the discussion and conclusions.Take notice of all reviewer suggestions.

We look forward to receiving your revised manuscript.

Kind regards,

Ratna Dwi Wulandari, Dr

Guest Editor

PLOS ONE

Journal Requirements:

2. Please ensure that you have specified (1) whether consent was informed and (2) what type you obtained (for instance, written or verbal, and if verbal, how it was documented and witnessed). If your study included minors, state whether you obtained consent from parents or guardians. If the need for consent was waived by the ethics committee, please include this information.

3. Please could you clarify within the ethics statement whether the 'Information Governance Review Panel of Swansea University' is the same committee as the Research Ethics - Swansea University.

This work was supported by the Medical Research Council (MRC), grant no. 232

MR/S027750/1. FJ is supported by a MRC/University of Manchester Skills 233

Development Fellowship (grant number MR/R016615). The funders had no role in 234

study design, data collection and analysis, decision to publish, or preparation of the 235

manuscript.

However, funding information should not appear in the Acknowledgments section or other areas of your manuscript. We will only publish funding information present in the Funding Statement section of the online submission form. 

This work was supported by the Medical Research Council (MRC), grant no. MR/S027750/1. FJ is supported by a MRC/University of Manchester Skills Development Fellowship (grant number MR/R016615). The funders had no role in study design, data collection and analysis, decision to publish, or preparation of the manuscript. 

Additional Editor Comments:

This post is exciting and well-written. However, here are some areas that need improvement.

Background.

The author explains that the most frequently used approach so far in multi-morbidity investigations is statistical modeling and unsupervised machine learning. Before going into the explanation of hypergraphs, the author should first explain what are the weaknesses of statistical modeling and unsupervised machine learning, so it is necessary to do research with hypergraphs.

The author also needs to add examples of the advantages of using hypergraphs based on previous research.

Writers should write down the research objectives clearly.

Method

Page 2 line 46, the first sentence is confusing, because the author wrote down the cohort used for this research has been explained before. It will be easier for readers to understand if the writer starts by explaining why they use cohort data. The author also has not explained well how to get cohort data. How does the author control the quality of the data?

There should be a more detailed explanation regarding the utilization of health resources.

Discussion

The author needs to elaborate on more literature and previous research results so that the discussion is more in-depth and interesting.

Try doing the ending separately.

Reviewers' comments:

Reviewer's Responses to Questions

**Comments to the Author**

1. Is the manuscript technically sound, and do the data support the conclusions?

Reviewer #1: Yes

2. Has the statistical analysis been performed appropriately and rigorously? 

Reviewer #1: Yes

3. Have the authors made all data underlying the findings in their manuscript fully available?

Reviewer #1: No

4. Is the manuscript presented in an intelligible fashion and written in standard English?

Reviewer #1: Yes

5. Review Comments to the Author

Reviewer #1: The authors clearly showed different approaches that have been used to analysed multi-morbidity outcome data citing examples. The further explain how different and noble their methods can be also adopted to analyse a different dimension of such multi-morbid outcome data.

For table 1 it will be best if the diseases are sorted either in ascending or descending order based on the percentages. The current arrangement does not seem to follow any reason order.

Given that hypertension seems to be the most prevailing condition, perhaps, the authors should also consider in future work a similar approach where they focus attention on multimorbidity among patients with hypertension. This will highlight the most important co-morbid conditions related to hypertension

6. PLOS authors have the option to publish the peer review history of their article (what does this mean?). If published, this will include your full peer review and any attached files.

Reviewer #1: No

---

## [Author Response · Author response to Decision Letter 0]

25 Jan 2023

Thank you for reviewing our paper. We have attached a full response to editor and referee comments to this submission as a file attachment. Please see that file for a point by point response to referee and editor comments.

---

## [Decision Letter · Decision Letter 1]

17 Apr 2023

PONE-D-22-20775R1Using hypergraphs to quantify importance of sets of diseases by healthcare resource utilisation: A retrospective cohort studyPLOS ONE

Dear Dr. Rafferty,

Thank you for submitting your manuscript to PLOS ONE. After careful consideration, we feel that it has merit but does not fully meet PLOS ONE’s publication criteria as it currently stands. Therefore, we invite you to submit a revised version of the manuscript that addresses the points raised during the review process.

We look forward to receiving your revised manuscript.

Kind regards,

Ratna Dwi Wulandari, Dr

Guest Editor

PLOS ONE

Journal Requirements:

Additional Editor Comments:

Here are some suggestions for improving your manuscript based on the second reviewer's comments.

This paper deals with an important problem. How do we deal with the thorny problem of multimorbidity - it is a high dimensional problem. They present a dimension reduction technique, the use of hypergraphs. The data used is suitable for addressing this question. It is good that code is provided.

However, the problem is that, after reading the paper, I am none the wiser as to the role of hypergraphs. I can't interpret figure 2 and am not really sure what the algorithm is doing or what the significance of that is. I cannot bridge the gap between the technical explanation and what is presented. The method is so unfamiliar, and it is the method that is the novel thing here, that I think the authors need to take us through it, almost in the form of a tutorial. I appreciate that this is challenging, but I think we need much more help as readers to understand what is being proposed. Could the authors bring in simulated examples, or examples from other fields?

1. I am afraid that I cannot follow what is meant in the background in the paragraph starting "Research". For example what does "This symmetry allows one to construct the dual hypergraph" mean? This approach is much less familiar to readers than is regression modelling. I think we need some examples and we need to get some kind of intuition or feel for what the algorithm is doing and what it means.

2. I don't think the term "important" is helpful here either when discussing prevalence of HRUs or their combination. Could it instead talk about the most common and most costly? For their combination some other term would be useful. Maybe most "prominent"?

3. The text on Figure 2 is too small. More importantly, we need some text to talk us through the figure to help explain it.

Reviewers' comments:

Reviewer's Responses to Questions

**Comments to the Author**

1. If the authors have adequately addressed your comments raised in a previous round of review and you feel that this manuscript is now acceptable for publication, you may indicate that here to bypass the “Comments to the Author” section, enter your conflict of interest statement in the “Confidential to Editor” section, and submit your "Accept" recommendation.

Reviewer #2: (No Response)

2. Is the manuscript technically sound, and do the data support the conclusions?

Reviewer #2: Partly

3. Has the statistical analysis been performed appropriately and rigorously? 

Reviewer #2: I Don't Know

4. Have the authors made all data underlying the findings in their manuscript fully available?

Reviewer #2: No

5. Is the manuscript presented in an intelligible fashion and written in standard English?

Reviewer #2: Yes

6. Review Comments to the Author

Reviewer #2: This paper deals with an important problem. How do we deal with the thorny problem of multimorbidity - it is a high dimensional problem. They present a dimension reduction technique, the use of hypergraphs. The data used is suitable for addressing this question. It is good that code is provided.

However, the problem is that, after reading the paper, I am none the wiser as to the role of hypergraphs. I can't interpret figure 2 and am not really sure what the algorithm is doing or what the significance of that is. I cannot bridge the gap between the technical explanation and what is presented. The method is so unfamiliar, and it is the method that is the novel thing here, that I think the authors need to take us through it, almost in the form of a tutorial. I appreciate that this is challenging, but I think we need much more help as readers to understand what is being proposed. Could the authors bring in simulated examples, or examples from other fields?

1. I am afraid that I cannot follow what is meant in the background in the paragraph starting "Research". For example what does "This symmetry allows one to construct the dual hypergraph" mean? This approach is much less familiar to readers than is regression modelling. I think we need some examples and we need to get some kind of intuition or feel for what the algorithm is doing and what it means.

2. I don't think the term "important" is helpful here either when discussing prevalence of HRUs or their combination. Could it instead talk about the most common and most costly? For their combination some other term would be useful. Maybe most "prominent"?

3. The text on Figure 2 is too small. More importantly, we need some text to talk us through the figure to help explain it.

7. PLOS authors have the option to publish the peer review history of their article (what does this mean?). If published, this will include your full peer review and any attached files.

Reviewer #2: **Yes: **David A McAllister

---

## [Author Response · Author response to Decision Letter 1]

2 Jun 2023

Please find a response to the reviewers comments as an attached file.

---

## [Decision Letter · Decision Letter 2]

21 Jun 2023

PONE-D-22-20775R2Using hypergraphs to quantify importance of sets of diseases by healthcare resource utilisation: A retrospective cohort studyPLOS ONE

Dear Dr. Rafferty,

Thank you for submitting your manuscript to PLOS ONE. After careful consideration, we feel that it has merit but does not fully meet PLOS ONE’s publication criteria as it currently stands. Therefore, we invite you to submit a revised version of the manuscript that addresses the points raised during the review process.

We look forward to receiving your revised manuscript.

Kind regards,

Ratna Dwi Wulandari, Dr

Guest Editor

PLOS ONE

Journal Requirements:

Additional Editor Comments:

The reviewers still provide some comments in this third round, so the writer needs to improve the manuscript again, following the reviewer's comments.

Reviewers' comments:

Reviewer's Responses to Questions

**Comments to the Author**

1. If the authors have adequately addressed your comments raised in a previous round of review and you feel that this manuscript is now acceptable for publication, you may indicate that here to bypass the “Comments to the Author” section, enter your conflict of interest statement in the “Confidential to Editor” section, and submit your "Accept" recommendation.

Reviewer #2: (No Response)

2. Is the manuscript technically sound, and do the data support the conclusions?

Reviewer #2: Yes

3. Has the statistical analysis been performed appropriately and rigorously? 

Reviewer #2: Yes

4. Have the authors made all data underlying the findings in their manuscript fully available?

Reviewer #2: No

5. Is the manuscript presented in an intelligible fashion and written in standard English?

Reviewer #2: No

6. Review Comments to the Author

Reviewer #2: Thanks for highlighting the Rafferty et al paper (ref 24) using hypergraphs for multimorbidity. The explanations in that were great. It also gave a nice intuition for the concept of the centrality metric and the example of Google's algorithm was very helpful. It would be good to make that paper more prominent in the background.

I agree that in the light of the Rafferty et al paper it is not necessary to write a tutorial-type paper here. However, I think this paper still needs a bit more to give the reader an intuitive understand both networks and centrality as a measure. Something analogous to the papers that reported network meta-analyses before this became a common approach.

I think the paper needs something like the following in the introduction (not my area so the following likely to be wrong, but to give the tone):-

- Describing and summarising multimorbidity is challenging because ...

- approaches that have been used include simple counts, weighted counts, clustering algorithms, and multi-state models (as per current intro)

- Network based approaches are promising because in describing sets of conditions, they allow us to account for the prevalence of individual conditions as well as their relations to other conditions, with varying degrees of commonness. We used a network approach in a previous original article and tutorial paper (24)

- we used hypergraphs for that article rather than simpler binary graphs because XXX and generated summary measures of XXX called centrality. Centrality measures are good because XXX [some intuitive description in the intro]

- Prevalence, is only one way to calculate ... it is possible to weight by any continuous characteristic, and doing so can give different insights into conditions and their relations.

- We now do so for healthcare resource utilisation, which like prevalence is a very important ...

As an aside, I think having a background and an introduction is overkill. It would be better to collapse these into a single section.

7. PLOS authors have the option to publish the peer review history of their article (what does this mean?). If published, this will include your full peer review and any attached files.

Reviewer #2: **Yes: **David A McAllister

---

## [Author Response · Author response to Decision Letter 2]

10 Jul 2023

Thank you for reviewing our paper. We have attached a full response to editor and referee comments to this submission as a file attachment. Please see that file for a point by point response to referee and editor comments.

---

## [Editor Report · Decision Letter 3]

21 Nov 2023

Using hypergraphs to quantify importance of sets of diseases by healthcare resource utilisation: A retrospective cohort study

PONE-D-22-20775R3

Dear Dr. Rafferty,

We’re pleased to inform you that your manuscript has been judged scientifically suitable for publication and will be formally accepted for publication once it meets all outstanding technical requirements.

Kind regards,

Ratna Dwi Wulandari, Dr

Guest Editor

PLOS ONE
---

## [Editor Report · Acceptance letter]

7 Dec 2023

PONE-D-22-20775R3 

Using hypergraphs to quantify importance of sets of diseases by healthcare resource utilisation: A retrospective cohort study 

Dear Dr. Rafferty:

I'm pleased to inform you that your manuscript has been deemed suitable for publication in PLOS ONE. Congratulations! Your manuscript is now with our production department. 

Kind regards, 

on behalf of

Prof. Ratna Dwi Wulandari 

Guest Editor

PLOS ONE